# A Boo(n) for Evaluating Architecture Performance

**Ondrej Bajgar** [1]   **Rudolf Kadlec** [1]   **Jan Kleindienst** [1]

## Abstract

We point out important problems with the common practice of using the best single model performance for comparing deep learning architectures, and we propose a method that corrects these flaws. Each time a model is trained, one gets a different result due to random factors in the training process, which include random parameter initialization and random data shuffling. Reporting the best single model performance does not appropriately address this stochasticity. We propose a normalized expected best-out-of-$n$ performance ($\text{Boo}_n$) as a way to correct these problems.

## 1. Introduction

Replicating results in deep learning research is often hard. This harms their usefulness to industry, leads to a waste of effort by other researchers, and limits the scientific value of such results.

One reason is that many papers provide information insufficient for replication. Details of the experimental setup can significantly influence the results (Henderson et al., 2018; Fokkens et al., 2013; Raeder et al., 2010), so the details should be provided at least in appendices, ideally alongside the source code, as was strongly emphasized e.g. by Ince et al. (2012).

However, an important second factor hinders replicability: most deep learning training methods are inherently stochastic. This randomness usually comes from *random* data ordering in *stochastic* gradient descent and from *random* parameter initialization, though there can be additional sources of randomness such as dropout or gradient noise. Consequently, even if we fix the model architecture and the experimental setup (including the hyperparameters), we obtain a different result each time we run an experiment. Statistical techniques are needed to handle this variability. However,

in deep learning research, they are heavily underused. What is usually done instead?

Most empirical deep learning papers simply report the performance of the best single model (sometimes calling it just "single model" performance). We will later show this is the case at least for some sub-domains. Given the result stochasticity, such method is statistically flawed. The best model performance is not robust under experiment replication, and its expected value improves with an increasing number of experiments performed, among other problems. Since many deep learning publications largely ignore these issues, we dedicate the first part of this article to explaining them in some detail, and later run experiments to quantify them.

Appropriate statistical techniques are hence necessary for evaluating (and comparing) the performance of machine learning (ML) architectures. Some well-developed methods exist for such comparisons (a great introduction is given for instance by Cohen (1995)). However, most existing methods focus on comparing the *mean* performance. This may be one of the reasons why statistical methods are being underused, since mean may be unattractive to researchers in certain situations.

There are multiple possible reasons for this. The one that we do consider sound[1] is that when deploying models in practice, it is often natural to train multiple instances of a model and then deploy the best one to production based on a validation set evaluation.[2] Underperforming models can be discarded, so the final deployed model does come from

---

[1]  IBM Watson, Prague AI Research & Development Lab. RK has since moved to Deepmind. Correspondence to: Ondrej Bajgar < ondrej@bajgar.org >.

Originally published in the *Proceedings of the 35th International Conference on Machine Learning*, Stockholm, Sweden, PMLR 80, 2018. Copyright 2018 by the author(s).

---

[1]Other reasons why researchers resort to the best performance as opposed to the mean may come from the current highly competitive atmosphere in the field with (possibly excessive) focus on performance on standard datasets (see Church (2017) or Sculley et al. (2018) for further discussion), which may motivate researchers to publish only their best results. Also, statistically sound estimation of performance does require repeatedly re-running experiments, which does incur additional cost, which researchers may prefer to invest in additional model tuning, especially in the present situation where reviewers seem not to require statistically sound evaluation of models and on the other hand may favour high-performing models. Of course, these motives should instead give place to effort to do good science, as opposed to a race on standard benchmarks.

[2]In some applications there is focus on speed of training and on reducing computational costs – there it does make sense to focus on the performance of the typical model as opposed to the best out of $n$, so the use of mean or median is appropriate.

the higher tier of the model performance population, and the use of mean may be inappropriate.

Hence, rather than to completely abandon reporting the performance of the best model, we propose a way to fix its flaws. We do this by estimating the expected best-out-of-$n$ ($\mathrm{Boo}_n$) performance by running more than $n$ experiments, which gives the estimate statistical validity if a sufficient number of experiments are run. We discuss how this measure relates to the performance distribution of the model, and we also give a method to empirically estimate $\mathrm{Boo}_n$.

The paper proceeds as follows: First, we give a high-level explanation of why reporting performance of the best single model is problematic. We also give some evidence that it is widely used in the deep learning community, which is why this explanation may be needed. We proceed by presenting $\mathrm{Boo}_n$ as a way to fix the above problems. We then give some experimental evidence for the flaws of best-single-model reporting and show that $\mathrm{Boo}_n$ does not suffer from them. We wrap up by discussing the place of $\mathrm{Boo}_n$ in a ML researcher's toolbox alongside traditional measures such as mean or median.

## 2. Best Single Model Performance

In articles presenting new deep learning architectures, the performance is often reported as the score of the "best single model" or simply "single model". In practice, this usually means that the researchers train multiple instances of the proposed architecture (often with different sets of hyperparameters), evaluate these instances on some validation set, and select the best-performing model. This best model is evaluated on a test set, and the resulting test score is then reported as the metric characterizing the architecture and used for comparing it to previous models. If the score is better than those reported in previous work, the architecture is presented as superior. This practice results in several issues:

**Population variance**  Since results of experiments are stochastic, the performance of a single model is just a single instance drawn from a possibly disparate population. If others train the model on their own, they get another sample from the architecture's performance distribution, which may substantially differ from the one listed in the original paper. Such paper thus gives insufficient information about what to expect from the new architecture, which should be one of the article's main purposes.

One may object that the result published in the paper is not chosen from the population at random – it is selected using a validation result. However, the correlation between the validation and test results is generally imperfect; in fact, in some of our experiments, it is almost zero, as we show in Section 4. Furthermore, if we indeed do have a strong

correlation, we get another problem:

**Expectation of best result increases with the number of experiments**  Simply put, the more samples from a population we draw, the more extreme the best among them is likely to be. In other words, the expected value of the best result depends on the number of experiments that the researchers run. There are three closely related problems with this: Firstly, this makes the number of experiments run an important explanatory variable; however, this variable is usually unreported, which is a severe methodological flaw in itself. It also leads to the second problem: since each research team runs a different number of experiments, the results are not directly comparable. Thirdly, this motivates researchers to run more experiments and gives an advantage to those who are able to do so. This pushes publishing quantitative results towards a race in computational power rather than a fair comparison of architectures themselves.

**Best model performance is not a meaningful characteristic of the performance distribution**  Even if we knew the underlying theoretical performance distribution – that is, if we had perfect information about the architecture's performance – it would not be clear what we would mean by "best model performance" without specifying the size of the pool from which we are choosing the best model. Imagine some architecture having a Gaussian performance distribution. Asking what is the best possible performance does not make sense in such a case, since the support of the distribution is unbounded. Even for capped metrics such as accuracy, where the performance distribution necessarily has bounded support, the best (possible) model[3] may be so unlikely, that it would be of no practical importance. Hence, best model performance is not a meaningful characteristic of the performance distribution.

**Generality / Falsifiability**  Finally, there is the question of what the authors are trying to express. Using "best single model performance", they are essentially claiming: "There once existed an instance of our model that once achieved a result X on dataset Y". Such fact is not of that much interest to the scientific community, which would rather need to know how the architecture behaves *generally*. Relatedly, a frequently given characteristic of science is *falsifiability* of theories (Popper, 1959). A theory claiming that there are invisible unicorns running among us is not science, since we cannot think of any potential empirical evidence that could prove the theory false. Similarly, any number of replication experiments that produce substantially worse results cannot prove the above performance claim wrong. If, for instance, a confidence interval were given, replications could

---

[3]In the sense of validation performance being at or near the supremum of the validation performance distribution's support.

very quickly show the published result at least extremely implausible, if not false.

We will quantify the former two problems for two concrete architectures in Section 4.

### 2.1. Prevalence

Despite all these problems, reporting the performance of the best model is still the main way of reporting results in some areas of ML, especially in empirically oriented deep learning papers, and, alarmingly, such practice seems to be tolerated by reviewers even at prime conferences. For instance, what concerns models published on the popular Children's Book Test dataset for reading comprehension (on which we run experiments later), none of the (more than ten) papers used any form of statistical testing or confidence intervals, and most reported only performance of the best single model without even mentioning the total number of experiments run. These include papers published at NIPS (Hermann et al., 2015), ICLR (Hill et al., 2016; Munkhdalai & Yu, 2017), ACL (Chen et al., 2016; Dhingra et al., 2017; Cui et al., 2017), or EMNLP (Trischler et al., 2016).

The same is true for the recently popular SQuAD dataset: for instance, none of the four papers (Yang et al., 2017; Wang & Jiang, 2017; Seo et al., 2017; Xiong et al., 2017) that published results on this dataset at ICLR 2017 has used any statistical testing or confidence intervals nor published mean (or otherwise aggregated) results across multiple runs.

Let us look more generally at the example of ICLR 2017 (chosen as a deep-learning-focused conference featuring many empirical results – as a rough guide, 174 out of 194 ICLR papers have "experiment" in a (sub-)section heading). Only 11 papers mention terms related to hypothesis testing[4], and 11 contain the string "confidence interval". Further details can be found in Appendix B.

While this is a rough and limited survey, it does suggest that while deep learning research is to a large extent an empirical science, statistical methods are often underused.

## 3. Expected Best-out-of-$n$ (Boo$_n$) Performance

The issues outlined above point to desiderata for a more suitable method of reporting an architecture's performance. It should provide information about general behaviour of the architecture under specified conditions, well characterizing

the associated random performance distribution. It should also be invariant under the number of experiments run and be as robust as possible to random noise.

Given these requirements, traditional statistical measures, such as mean or median, probably come to mind of many readers. They do indeed fix the above issues; still, they express only the performance of a typical member of the population. However, in many ML applications, it may be the best model from a pool that is of interest. When practitioners are choosing a model for deployment, they train multiple models and deploy the best-performing one [5]. This gives some justification to reporting the performance of the best model and gives us a reason to attempt to fix its problems rather than completely dismiss it. Such corrected best-model measure would be more informative than mean or median in these outlined situations.

A natural way to improve comparability between models, each evaluated in a different number of experiments, is to normalize the results to the expected result if the number of experiments were the same, say $n$, which can be easily estimated if we run $m$ experiments, $m \geq n$. The greater the number of experiments $m$, the more robust the estimate of the expected best, which also helps us eliminate the problem of statistical robustness. We are proposing the expected best-out-of-$n$ performance, Boo$_n$, to be used where the performance of the best model from a pool seems as an appropriate measure.

Let us first examine how the expected best-out-of-$n$ (Boo$_n$) performance relates to the (theoretical) performance distribution we are trying to characterize; we will then proceed with empirical estimation, which is of value in practice. The calculations are not particularly innovative from statistical point of view and are close to many standard results from the field of Order Statistics (see for instance Arnold et al. (2008) for more context).

### 3.1. Boo$_n$ of a probability distribution

Showing how to calculate Boo$_n$ of a known theoretical probability distribution will serve two purposes: Firstly, since we are proposing Boo$_n$ as a way to characterize the performance distribution, this will make the relation between Boo$_n$ and the performance distribution explicit. Secondly, in some cases we may be able to make an assumption about the family to which the theoretical distribution belongs (e.g.

---

[4]This was checked by searching for any of the following strings: "hypothesis test", "p-value", "t-test" "confidence level", "significance level", "ANOVA", "analysis of variance", "Wilcoxon", "sign test".

[5]This would usually be the case when a model is trained once and then deployed for longer-term usage, which may be the case for instance for Machine Translation systems. In other cases, when it is practical to train only as single model instance due to hardware constraints (either because training is extremely costly, or because it needs to be done repeatedly, e.g. for individual customers), we may indeed be interested in a typical model and hence in mean or median performance.

we could assume it is approximately Gaussian). The analytic calculation below will allow us to leverage this information when empirically estimating $\mathrm{Boo}_n$ by deducing a parametric estimator, which may be especially useful when our sample size $m$ is small thanks to its lower variance due to added prior information.

### 3.1.1. SINGLE EVALUATION

Let us first look at the simpler case of validation performance (that is, the case where we are choosing the best model with respect to the metric we are reporting) as it is easier to grasp: How do we calculate an expected best $\overline{\mathrm{Boo}_n}(\mathfrak{P})$[6] of independent identically distributed (i.i.d.) random variables $X_1, ..., X_n$ with probability distribution $\mathfrak{P}$ (the *performance distribution* of an architecture) with a probability density function (p.d.f.) $f$ and a cumulative distribution function (c.d.f.) $F$? In the case where best means maximal (minimum can be calculated similarly), the maximum $\max\{X_1, ..., X_n\}$ has a c.d.f. equal to

$$F_{\max}(x) = \mathbb{P}[\max\{X_1, ..., X_n\} \leq x] =$$
$$\mathbb{P}[X_1 \leq x, ..., X_n \leq x] = F^n(x) \quad (1)$$

using the independence of the $X_i$s in the last step.

In the case of a **continuous** distribution, we can obtain the p.d.f. of the maximum by simply differentiating with respect to x:

$$f_{\max}(x) = \frac{d}{dx}F_{\max}(x) = nf(x)F^{n-1}(x)$$

Using the p.d.f., we can now calculate the expected value of the maximum as

$$\overline{\mathrm{Boo}_n}(\mathfrak{P}) = \int_{-\infty}^{\infty} x f_{\max}(x)dx = \int_{-\infty}^{\infty} x n f(x) F^{n-1}(x)dx \quad (2)$$

We can get a precise numerical estimate of the above integrals in any major numerical computation package such as *numpy*. For illustration, for the standard normal distribution we have $\mathrm{Boo}_5(\mathcal{N}(0,1)) \approx 1.163$, $\mathrm{Boo}_{10}(\mathcal{N}(0,1)) \approx 1.539$. More generally, $\overline{\mathrm{Boo}_n}(\mathcal{N}(\mu, \sigma^2))$ can then be expressed as $\mu + \sigma\overline{\mathrm{Boo}_n}(\mathcal{N}(0,1))$. Thanks to this form we can get numerical estimates of $\overline{\mathrm{Boo}_n}(\mathcal{N}(\mu, \sigma^2))$ just by estimating the two usual parameters of the Gaussian, $\overline{\mathrm{Boo}_n}(\mathcal{N}(0,1))$ becoming just a constant coefficient if we fix $n$. The full details of calculation for the Gaussian distribution can be found in Appendix A.

In the case of a **discrete** performance distribution, which will be useful for empirical estimation below, we get a prob-

---

[6]Using the overline to distinguish from the double validation-test evaluation case later.

ability mass function

$$\mathbb{P}[\max\{X_1, \ldots, X_n\} = m] =$$
$$\mathbb{P}[\max\{X_1, \ldots, X_n\} \leq m] - \mathbb{P}[\max\{X_1, \ldots, X_n\} < m] \quad (3)$$

so if $p_j$ is the probability weight associated with value $x_j$, i.e. $\mathbb{P}[X_i = x_j] = p_j$ for all $i$, this gives us

$$\overline{\mathrm{Boo}_n}(\mathfrak{P}) = \sum_i \left( \left( \sum_{j:\ x_j \leq x_i} p_j \right)^n - \left( \sum_{j:\ x_j < x_i} p_j \right)^n \right) x_i \ . \quad (4)$$

### 3.1.2. VALIDATION-TEST EVALUATION

In the previous part, we were choosing the best model with respect to the metric whose expectation we were calculating. Hence, that method can be used to calculate the expected best validation performance of $n$ models. In practice, the best model is usually chosen with respect to the validation performance, while the primary interest is in the corresponding test performance. To calculate the expected test performance of the best-validation model, we need to substitute the direct value of $x$ in Equations 2 and 4, with the expectation of the test performance $X_{\mathrm{test}}$ conditional on the validation performance $x_{\mathrm{val}}$,

$$E_{tv}(x_{\mathrm{val}}) := \mathbb{E}[X_{\mathrm{test}}|X_{\mathrm{val}} = x_{\mathrm{val}}]$$

yielding an expression for the expected test performance of the best-validation model chosen from a pool of size $n$

$$\mathrm{Boo}_n(\mathfrak{P}) = \int E_{tv}(x_{\mathrm{val}})d\mathfrak{P}_{\mathrm{val}}^{\mathrm{best}}(x_{\mathrm{val}}) =$$
$$\int_{-\infty}^{\infty} E_{tv}(x_{\mathrm{val}})n f_{\mathrm{val}}(x_{\mathrm{val}})F_{\mathrm{val}}^{n-1}(x_{\mathrm{val}})dx_{\mathrm{val}} \quad (5)$$

where $\mathfrak{P}_{\mathrm{val}}^{\mathrm{best}}$ is the marginal probability distribution of the best-out-of-$n$ validation performance. Similar simple substitution can be done in the discrete case.

Expanding the expression for the bivariate Gaussian distribution having marginal test performance with mean $\mu_{\mathrm{test}}$, standard deviation $\sigma_{\mathrm{test}}$, and test-validation correlation $\rho$ as in Appendix A.2 yields a convenient expression

$$\mathrm{Boo}_n = \mu_{\mathrm{test}} + \rho\, \sigma_{\mathrm{test}}\, \overline{\mathrm{Boo}_n}(\mathcal{N}(0,1))\ , \quad (6)$$

which can again be used for parametric estimation.

### 3.2. Empirical estimation

We usually do not know the exact performance distribution of the model; we only have samples from this distribution – the results of our experiments. In such case, we can estimate the expected maximum empirically, and in fact it is the

empirical estimates that are likely to be used in practice to compare models.

To get a non-parametric estimator, for which we do not make any assumption about the family of the performance distribution, we take the empirical distribution arising from our sample as an approximation of the architecture's true performance distribution, similarly to Bootstrap methods. The empirical performance distribution $\widehat{\mathfrak{P}}$ assigns a probability weight of $\frac{1}{m}$ to each of our $m$ samples. We approximate $\text{Boo}_n$ of the true performance distribution by $\text{Boo}_n$ of this empirical distribution. For the uniform empirical distribution, all the $p_i$ in Equation 4 are equal to $\frac{1}{m}$. Hence, if we rank our samples from worst-validation to best-validation as $(x_{i_1}^{\text{valid}}, x_{i_1}^{\text{test}}), \ldots, (x_{i_m}^{\text{valid}}, x_{i_m}^{\text{test}})$, we get

$$\widehat{\text{Boo}_n}\left((x_1^{\text{valid}}, x_1^{\text{test}}), \ldots, (x_m^{\text{valid}}, x_m^{\text{test}})\right) =$$
$$\sum_{j=1}^{m} \left(\left(\frac{j}{m}\right)^n - \left(\frac{j-1}{m}\right)^n\right) x_{i_j}^{\text{test}} \quad . \quad (7)$$

This is in fact a weighted average of the test results. In case of a tie in validation results, i.e. if $x_{i_{j+1}}^{\text{valid}} = x_{i_{j+2}}^{\text{valid}} = \ldots = x_{i_{j+k}}^{\text{valid}}$, one should assign an equal weight of $\left(\left(\frac{j+1}{m}\right)^n - \left(\frac{j}{m}\right)^n + \cdots + \left(\frac{j+k}{m}\right)^n - \left(\frac{j+k-1}{m}\right)^n\right)/k = \left(\left(\frac{j+k}{m}\right)^n - \left(\frac{j}{m}\right)^n\right)/k$ to the corresponding test samples[7].

This estimator does not make any assumption about the performance distribution from which our observations are drawn. If we do use such an assumption (e.g. we know that the performance distribution of our architecture usually belongs to a certain family, e.g. Gaussian), we can add information to our estimator and possibly get an even better estimate (i.e. one with lower sampling error). For the Gaussian distribution, we can use the standard estimators of the parameters in Equation 6 to get a parametric estimator

$$\widehat{\mu_{\text{test}}} + \widehat{\rho}\, \widehat{\sigma_{\text{test}}}\, \overline{\text{Boo}_n}\left(\mathcal{N}(0,1)\right)$$

where $\widehat{\mu_{\text{test}}}$, $\widehat{\rho}$ and $\widehat{\sigma_{\text{test}}}$ are standard estimators of mean, correlation, and standard deviation respectively. A similar parametric estimator could be calculated for other distributions.

### 3.3. Choice of $n$

$\text{Boo}_n$ eliminates the problem of dependence on the number of experiments run, $m$. However, we still need to choose $n$, the number of experiments to which we normalize. This is similar to the choice one is facing when using a quantile – should one use the 75% one, the 95% one, or some other?

The choice of $n$ most useful to a reader is when $n$ is the number of candidate models that a practitioner would train

---

[7]You can get an optimized Python implementation of the non-parametric estimator via `pip install boon`.

before choosing the best one for some target application. Such number will differ domain to domain and will heavily depend on the computational cost of training the specific architecture on the specific domain. The $n$ of interest may differ for each reader – ideally, researchers should characterize the architecture's performance distribution as fully as possible and the readers may be able to deduce the value of $\text{Boo}_n$ for whichever $n$ they choose (up to a limit).

Leaving an additional degree of freedom in the choice of metric creates a risk of cherry picking. However, in many areas of machine learning, there already are many available metrics. Still, the main reporting metric seems to quickly converge on each task. The first published paper makes a choice; the subsequent ones usually follow suit (else they risk a suspicion that the architecture is not competitive on the previous metric). We believe similar convergence is likely for $\text{Boo}_n$ on each task.

In our experiments, we decided to use $n = 5$ – the AS Reader model which we use for our experiments takes about 2 hours to train on a single GPU, so someone replicating the $\text{Boo}_5$ performance could expect to achieve it overnight, which seems a to be a reasonable requirement.

### 3.4. Accounting for estimator uncertainty

Even $\text{Boo}_n$ is just a single number whose estimate can be noisy. Hence, with $\text{Boo}_n$, as well as with mean and other ways of aggregating results of a wider population, we should always use appropriate statistical methods when trying to compare the quantitative performance of a new model against a baseline. This can be done using significance testing (such as the $t$-test), or with the help of confidence intervals, which seems to be the method preferred by a significant part of the scientific community (e.g. Gardner & Altman (1986) or Berrar & Lozano (2013)), since it allows us to disentangle the effect size from the uncertainty associated with noise and sample size.

For some theoretical distributions, there exist ways to calculate the hypothesis test or confidence interval analytically (e.g. using the t-test or standard normal quantiles for the Gaussian). However, in cases where the family of the performance distribution or of the estimator is not known, we need to resort to computational methods - usually Monte Carlo (if we do know at least the family of the performance distribution) or the Bootstrap (Efron, 1979) (if we do not). A brief description of how to calculate confidence intervals using the Bootstrap is provided in the Appendix.

## 4. Experiment Results

*Note: The data and code for their analysis can be found at* `http://gitlab.com/obajgar/boon`, *along with Python functions you can use to calculate $\text{Boo}_n$.*

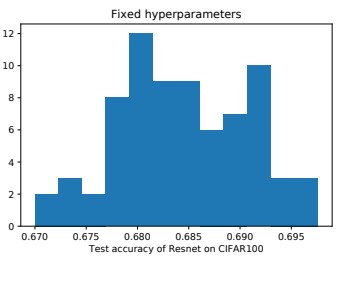
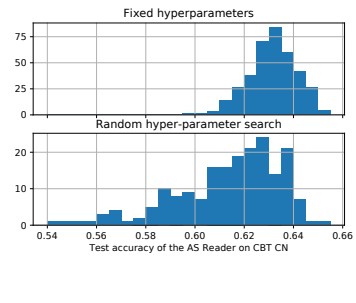
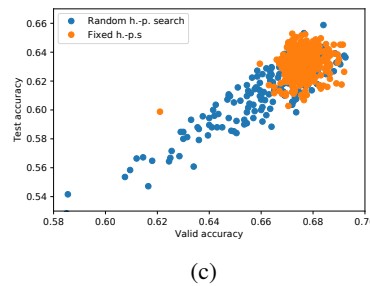

(a)  (b)  (c)

Figure 1: (a) & (b) Distribution of test accuracies of 74 instances of Resnet evaluated on CIFAR100, 370 instances of the AS Reader model with fixed hyperparameters and 197 with random hyperparameters trained and evaluated on CBT Common Nouns. (c) Relationship between the test and validation accuracies of the AS Reader for random hyperparameter search and for fixed hyperparameters.

We have run several experiments to quantify the scope of the problems outlined in Section 2. We just briefly summarize the main results here for illustration; a more detailed description of the experiments and analysis in the form of an iPython notebook can be found in the Gitlab repository or in Appendix C.

**Performance variation**  To estimate the random variation of results, we repeatedly[8] trained models from two domains of deep learning: the ResNet (He et al., 2016) on the CIFAR-100 dataset (Krizhevsky & Hinton, 2009) to represent Image Recognition and the Attention Sum Reader (AS Reader) (Kadlec et al., 2016) on the Children's Book Test Common Nouns (CBT CN) (Hill et al., 2016) to represent Reading Comprehension. Each of these trainings generated a pair of a validation and test performances. The resulting empirical performance distributions are illustrated in Figure 1.

If we fix all hyperparameters, the interquartile ranges of the models' accuracies are $0.98(\pm0.09)\%$[9] and $1.20(\pm0.12)\%$ (absolute). This is comparable to the median differences between published results on these datasets: $0.86\%$ and $1.15\%$ respectively[10]. Hence, random variation in performance cannot be considered negligible as is now often done. Furthermore, if we allow the hyperparameters to vary (in our case by random search), the result variance further increases, which further amplifies the outlined effects. In the

case of the AS Reader the interquartile range increased to $2.9(\pm0.3)\%$ when we randomly picked hyperparameters from a range applicable to training the model on a single GPU. However, note that the problem of result incommensurability due to hyperparameter optimization is *not* the focus of this work. The method that we present here is still applicable to the problem in the case of *random* hyperparameter sampling for which we include results, however we aim to compensate mainly for randomness due to parameter initialization and data shuffling – which is significant in itself, as we have just demonstrated.

Several other articles confirm significant variation in model performance due to different random seeds: e.g van den Berg et al. (2016) in Speech Recognition, Henderson et al. (2018) in Deep Reinforcement Learning, or Reimers & Gurevych (2017) in Named Entity Recognition. They all agree that reporting performance scores of single models is insufficient to characterize architecture performance.

**Estimator variance**  Figure 2 shows the $95\%$ confidence intervals of best single model results compared to the $\text{Boo}_5$ performance for a range of result-pool sizes $m$. This is shown for the cases of both strong and weak test-validation correlation. In both cases $\text{Boo}_5$ is significantly less noisy than the best-single-model result. In fact in the case of random hyperparameter search, $\text{Boo}_n$ shows even smaller variation than the mean (due to the negative skew of the performance distribution).

**Best-model performance improves with the number of experiments**  We also mentioned that if only the performance of the best model is reported, the more experiments are run, the better the expected result. Figure 2b illustrates that this effect can indeed be fairly strong, if the validation performance is a good predictor of the test performance, as is the case of the AS Reader with random hyperparameter

---

[8]Specifically, 74 times for Resnet, 370 times for the AS Reader with fixed hyperparameters, and 197 times for the AS Reader with random hyperparameters.

[9]$\pm$ standard deviation computed using 99999 Bootstrap samples.

[10]We looked at results of successive architectures evaluated on the two tasks as listed in (Huang et al., 2017; Munkhdalai & Yu, 2017). We sorted the results with respect to the test performance and then calculated the differences between successive models. From these we calculated the median. Full details can be found in the Gitlab repository.

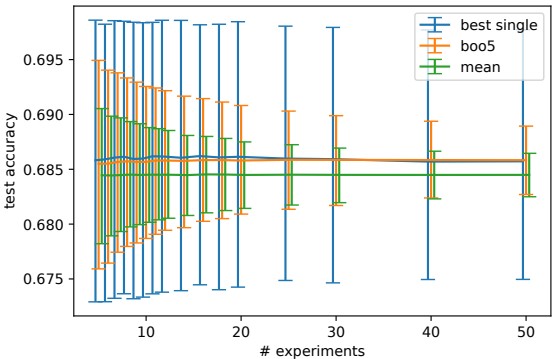

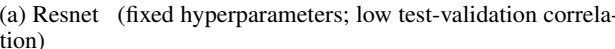

(a) Resnet (fixed hyperparameters; low test-validation correlation)

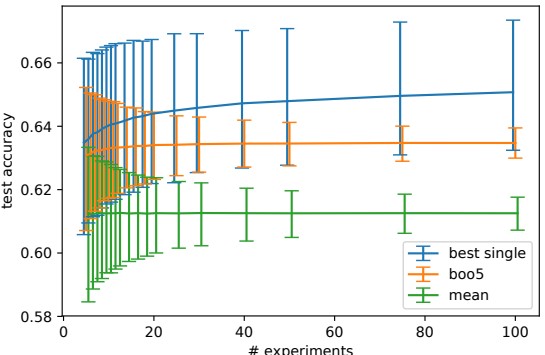

(b) AS Reader (random hyperparameter sampling; high test-validation correlation)

Figure 2: Averages and 95% confidence intervals of test performance for three ways of aggregating results of multiple experiments for various numbers of experiments run. Each confidence interval was constructed using smoothed[11] Bootstrap sampling from our pool of 75 for Resnet and 197 experiments for the AS Reader with fixed and random hyperparameters respectively. Since we strongly encourage researchers to provide confidence intervals for their results, we provide and overview of how to construct them using the Bootstrap in Appendix D.1.

search, where the expectation of the best single model performance increases from $61.3\%$ if we train it once, to $63.3\%$ if we train it 5 times, to $63.5\%$ for 20 times. This effect is nicely explained in more detail e.g. by Jensen & Cohen (2000). It gives a further argument for refraining from using this method and certainly also for publishing the number of experiments run, which is often not done. $\text{Boo}_n$ is not subject to this effect.

**Validation-test correlation** However, note that the assumption that validation performance is a good predictor of test performance is sometimes not true. In the two cases with fixed hyperparameters that we looked at, the Spearman correlation between validation and test results was only $0.10$ and $0.18$ respectively for the two models. The correlation significantly increases if we allow the hyperparameters to vary – to $0.83$ for the AS Reader. These results are also illustrated in Figure 1c. Larger validation sets are also likely to improve this correlation, which can be understood as the degree of generalization from validation to test. Note that the problem of increasing expected performance mentioned above is relevant only in the case of higher correlation between validation and test results. The effect becomes very strong in the case where the performance we are reporting is also used for choosing the best model, which emphasizes the need for honest separation of validation and test data.

---

[11]While $\text{Boo}_n$ and mean could be sampled using vanilla Bootstrap, best-validation result is influenced only by a single value from the sample and hence uses only few values from the upper tier of our result pool, which makes our pool size insufficient. Hence we use Gaussian kernel smoothing (Scott, 1992) to expand our result pool.

## 5. Discussion

$\text{Boo}_n$ does fix the main flaws of reporting the best single model performance. However, let us have a look at some of its limitations.

**Hyperparameter tuning** This work does not fully compensate for improved expected results due to hyperparameter tuning, nor was it its primary aim. $\text{Boo}_n$ is appropriate in the case of random hyperparameter sampling, where the performances in different runs are independent. However, this is not the case for more advanced hyperparameter optimization methods. The primary focus of this work was on tackling variability due to random initialization, data shuffling, and similar sources, which we have shown to be significant in itself. Compensation for more advanced hyperparameter tuning (and ensuring the comparability of models in that case) is certainly a worthwhile area for future research.

**Mean, median, and other alternatives** We do not claim our method to be strictly superior to traditional ways of aggregating results, such as mean or quantiles. However, we have outlined a case where using $\text{Boo}_n$ is justified – situations where a final model to be deployed can be chosen from a pool of trained candidates. In such case, $\text{Boo}_n$ is easily interpretable and more informative than a performance of a typical model, expressed by mean or median. Hence, we think $\text{Boo}_n$ is a useful addition to the methodological toolbox along existing methods.

**Just a single number**   $\text{Boo}_n$ is still just a single number whose ability to characterize the performance distribution is limited by its single dimension. Paper authors should try to characterise the performance distribution as fully as possible, which may involve a histogram, mean, standard deviation, ideally along a dataset containing the results of all experiments, from which an interested reader may be able to deduce whichever characteristic she finds interesting. Unfortunately, such characterization is usually lacking.

However, *alongside* this detailed characterization, describing an architecture's performance by a single number still has its appeal, especially for the purpose of comparison among architectures and choosing the best one according to some criterion (in fact, each quantitative score can be understood as a proxy for ordering architectures with respect to such criterion of interest, such as expected performance of the best model out of $n$). We have explained why, in some cases, $\text{Boo}_n$ may be useful for such purpose.

**Computational cost**   Some may deem $\text{Boo}_n$ impractical due to its requirement to train architectures many times, which may be very expensive in some cases. However, result stochasticity needs to be addressed to produce reliable results, and it is hard to imagine a general method to do so without repeated evaluation[12]. Researchers should focus on architectures which they can evaluate properly given their resources. However, the main target of our criticism is not projects whose resources are stretched by a single training; it is projects that do have the necessary resources for multiple evaluations but use them to produce better-looking results rather than results that are more informative and robust.

## 6. Conclusion

Reporting just the best single model performance is not statistically sound. This practice in machine learning research needs to change if the research is to have lasting value. Reviewers can play an important role in bringing this change.

Still, asking for the performance of a best model out of $n$ can have valid reasons. For the situations where the best-model performance is indeed a good metric, we are suggesting $\text{Boo}_n$ as a way to evaluate it properly.

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

# Appendix

## A. Boo$_n$ of the Gaussian Distribution

In this section we will calculate Boo$_n$ of the Gaussian distribution. This can serve as a basis for a parametric estimator of Boo$_n$, when we assume a performance distribution to be (approximately) Gaussian, which was the case of some of the performance distributions we have examined, for instance the AS Reader with fixed hyperparameters.

### A.1. Single evaluation dataset

In the simpler case in which the best model is chosen with respect to the same dataset on which the performance is then reported, we can substitute the p.d.f. and c.d.f. of the Gaussian distribution into Equation 2 to get

$$\overline{\text{Boo}_n}(\mathcal{N}(\mu, \sigma^2)) =$$
$$\int_{-\infty}^{\infty} x n \frac{1}{\sqrt{2\pi\sigma^2}} e^{\frac{(x-\mu)^2}{2\sigma^2}} \Phi^{n-1} \left( \frac{x-\mu}{\sigma} \right) dx \quad (8)$$

where $\Phi$ is the c.d.f. of a standard normal random variable. Substituting $z = \frac{x-\mu}{\sigma}$, $dx = \sigma dz$, yields

$$\int_{-\infty}^{\infty} n(\mu + \sigma z) \frac{1}{\sqrt{2\pi\sigma^2}} e^{\frac{z^2}{2}} \Phi^{n-1}(z) \sigma dz =$$
$$= \mu \int_{-\infty}^{\infty} n \frac{1}{\sqrt{2\pi}} e^{\frac{z^2}{2}} \Phi^{n-1}(z) dz +$$
$$+ \sigma \int_{-\infty}^{\infty} nz \frac{1}{\sqrt{2\pi}} e^{\frac{z^2}{2}} \Phi^{n-1}(z) dz = \mu + \sigma \overline{\text{Boo}_n}(\mathcal{N}(0,1))$$
$$(9)$$

(the first integrand has the form of the p.d.f. found above and hence integrates to one) so the expected maximum is neatly expressed in terms of a maximum of a standard normal and is linearly proportional to both the mean and the standard deviation. Once $n$ is fixed for comparison purposes, Boo$_n$ $(\mathcal{N}(0,1))$ is just a constant, e.g. Boo$_5$ $(\mathcal{N}(0,1)) \approx 1.163$, Boo$_{10}$ $(\mathcal{N}(0,1)) \approx 1.539$.

### A.2. Test-validation evaluation

Let us turn to the case of reporting the expected test set performance of a best-validation model. If we model the validation and test performances by a Bivariate Normal Distribution with valid-test correlation $\rho$, means $\mu_{\text{val}}, \mu_{\text{test}}$, and variances $\sigma_{\text{val}}^2, \sigma_{\text{test}}^2$, then given a validation performance $x_{\text{val}}$, the test performance is distributed normally with conditional expectation

$$E_{tv}(x_{\text{val}}) = \mu_{\text{test}} + \rho \frac{\sigma_{\text{test}}}{\sigma_{\text{val}}} (x_{\text{val}} - \mu_{\text{val}})$$

which gives

$$
\text{Boo}_n(\mathcal{N}(\mu, \sigma^2)) =
$$
$$
\int_{-\infty}^{\infty} \left( \mu_{\text{test}} + \rho \, \frac{\sigma_{\text{test}}}{\sigma_{\text{val}}} (x_{\text{val}} - \mu_{\text{val}}) \right) n \cdot
$$
$$
\cdot \frac{1}{\sqrt{2\pi\sigma_{\text{val}}^2}} e^{\frac{(x - \mu_{\text{val}})^2}{2\sigma_{\text{val}}^2}} \Phi^{n-1} \left( \frac{x - \mu_{\text{val}}}{\sigma_{\text{val}}} \right) dx \ . \quad (10)
$$

Using the same two tricks as above, this can be simplified to

$$
\text{Boo}_n(\mathcal{N}(\mu, \sigma^2)) = \mu_{\text{test}} + \rho \, \sigma_{\text{test}} \, \overline{\text{Boo}_n}(\mathcal{N}(0, 1)) \quad (11)
$$

where $\overline{\text{Boo}_n}(\mathcal{N}(0, 1))$ is the single-evaluation expected maximum of the standard normal distribution as defined above.

## B. Survey of ICLR 2017 Papers: Method

We downloaded the pdfs of all papers accepted to ICLR 2017[13], extracted text from them using the OpenSource Xpdf package[14] and then searched the resulting text documents using the grep command as follows.

Firstly, to roughly estimate the usage of experiments in the papers, we searched for the capitalized string "EXPERIMENT" in the documents, since all (sub-)section headings are capitalized in the ICLR format. This was matched in 174 documents. Further 6 contained the string "EVALUATION" yielding a total of 180 out of 194 papers containing one of the two strings, which suggests that many ICLR papers indeed have an empirical component, though our rough method is only very approximate.

We then searched for the string "confidence interval", which was matched in only 11 papers, and further 11 documents matched one of expressions related to hypothesis testing (curiously, a set completely disjoint from the "confidence interval" set). These terms were: "hypothesis test", "p-value", "t-test" "confidence level", "significance level", "ANOVA", "analysis of variance", "Wilcoxon", and "sign test". This may actually be only an upper bound since mentioning the term somewhere in the paper does not necessarily mean that the method was employed in the experimental procedure.

## C. Experiments: Details

*Note: The data and code for their analysis are available at* `http://gitlab.com/obajgar/boon`.

Here we provide further details of our experiments quantifying the extent of result stochasticity and the resulting effects.

### C.1. Models

To run our experiments we have chosen Open Source implementations[15] of models from two popular domains of deep learning, namely ResNet (He et al., 2016) on the CIFAR-100 dataset (Krizhevsky & Hinton, 2009) for Image Classification and the AS Reader (Kadlec et al., 2016) on the CBT CN dataset (Hill et al., 2016) for Reading Comprehension. We believe these two models are representative of models in their respective areas – Resnet is based on a deep convolutional network architecture as most recent models in machine vision, while the AS Reader is based on a bidirectional GRU network with attention, as is the case for many models in natural language processing.

### C.2. Data collection

To collect the data for our experiments, we repeatedly trained the two models. Each training instance had a different random parameter initialization and random data shuffling. We saved the model at least once per epoch and we then included the validation and test result from the best-validation evaluation as a single data point in our meta-dataset.

All training was done on Ubuntu 14.04 on a single GPU per training, either Nvidia Tesla K80 or GTX 1080.

#### C.2.1. RESNET

Resnet was trained with a single set of hyperparameters, the default ones for the above Open Source implementation. That means 5 residual units resulting in a 32-layer Resnet. The model was trained using the 0.9 momentum optimizer, with batch size 128, initial learning rate of 0.1 lowered to 0.01 after 40,000 steps and to 0.001 after 60,000 steps. Data augmentation included padding to 36x36 and then random cropping, horizontal flipping and per-image whitening. L2 regularization weight was set 0.002. The training ran for 300 epochs.

Training was done using Tensorflow 1.3.

### C.3. AS Reader

The AS Reader was trained in two different settings. Firstly 370 times with hyperparameters fixed to embedding dimension of 128 and 384 hidden dimensions in the GRU units,

---

[13]Downloaded from https://openreview.net/group?id=ICLR.cc//2017/conference from sections "Paper decision: Accept (Oral)" and "Paper decision: Accept (Poster)".

[14]http://www.xpdfreader.com/; we used version 4.00.01 on Debian Linux 9.2.

[15]The source code for Resnet can be found at `https://github.com/tensorflow/models/tree/master/research/resnet`; the code for the AS Reader at `https://github.com/rkadlec/asreader`.

with all other hyperparameters as used in the original AS Reader paper (Kadlec et al., 2016).

In the second setting, the hyperparameters for each training instance were chosen randomly from the following ranges: The batch size was chosen from the range $[16, 128]$, and the embedding size and hidden state size were each chosen from the range $[16, 512]$ with the $\log_2$ value of the parameter being distributed uniformly in the interval. The upper bounds of these intervals matched maximum training size feasible on our hardware using this implementation.

Training was done using Theano 0.9.0 and Blocks 0.2.0.

### C.4. Performance distribution results

Figure 1 plots the histograms of test performances of the evaluated models. The mean test accuracy for Resnet was $68.41\%$ with standard deviation of $0.67\%$ (absolute), the range was $67.31\% - 69.41\%$. For AS reader with fixed hyperparameters the mean was $63.16\%$ with standard deviation $0.94\%$ and range of $61.52\% - 64.60\%$. In the case of random hyperparameter search the mean was $61.26\%$, standard deviation $2.48\%$, and values ranged from $56.61\%$ to $64.01\%$.

In both cases with fixed hyperparameters the collected results are consistent with coming from a Gaussian distribution according to the Anderson-Darling test[16] (Anderson & Darling, 1954); the histograms also make it appear plausible that the performance distribution is approximately Gaussian. This is not the case for the random hyperparameter search where the distribution has a clear negative skew.

To put the above numbers into context, we also examined the margin of improvement of successive architectures published on the corresponding datasets, as listed in (Munkhdalai & Yu, 2017; Huang et al., 2017). We sorted the results with respect to the test performance and then calculated the differences between successive models. The median difference was for $0.86\%$ for CIFAR-100 and $1.15\%$ for CBT CN.

Note that the median differences are smaller than two standard deviations for each model. Two standard deviations from the mean approximately give the $95\%$ confidence interval for a Gaussian distribution – hence we could typically fit three successive published results within the width of one such confidence interval. The magnitude of the performance variation due to random initialization and data shuffling is therefore not negligible compared to the improvements in performance, which often hold an important place within articles in which they are presented. We hence think it is

---

[16]That is, despite the relatively large sample sizes, gaussianity cannot be ruled out at 0.05 significance level based on collected evidence.

inappropriate to completely ignore this random variation in evaluation protocols, which is currently the usual practice.

### C.5. Test-validation correlation

The best model is usually selected using validation performance [17]. This practice is based on the assumption that the validation accuracy is a reasonably good predictor of test accuracy. The results of our experiments, illustrated also in Figure 1, suggest that this assumption holds for performance variation due to hyperparameter choice. However, if we fix the hyperparameters, the correlation almost disappears. To some extent, this implies that selecting the best validation model means we are picking randomly with respect to the test performance. Since we are picking from a random test performance distribution, this further calls for better characterization of the distribution than a single instance drawn from it.

On the other hand if the correlation is strong, as seems to be the case if we do perform hyperparameter search, we face the second problem with reporting the best-validation performance:

### C.6. Effect of the number of experiments

If the validation performance is a good predictor of the test performance, then the more models we train the better the best-validation model is likely to be even on the test set since we are able to select models high up the right tail of the performance distribution. This effect has been described in more detail in (Jensen & Cohen, 2000), though with focus on induction algorithms; here we present an estimate of its effect in the case of Resnet and AS Reader.

To test this effect we took the pool of trained models. For each $m$ in the range from 1 to 50 (or 100 for the AS Reader), we randomly sampled $100,000$ samples of size $m$ from the pool, and selected the best-validation model from each sample. The mean test performance across the $100,000$ samples for each $m$ is plotted in Figure 2.

The results show that when there is suitable correlation between validation and test performances, increasing the number of experiments does increase the expected performance of the best-validation model. This makes the number of experiments an important explanatory variable, which, however, usually goes unreported. Furthermore, it makes results reported by different research teams not directly comparable. Finally, it gives an advantage to those that can run more experiments. We believe that this again makes the practice of reporting the performance of the best single model unsuitable.

---

[17]Or at least should be.

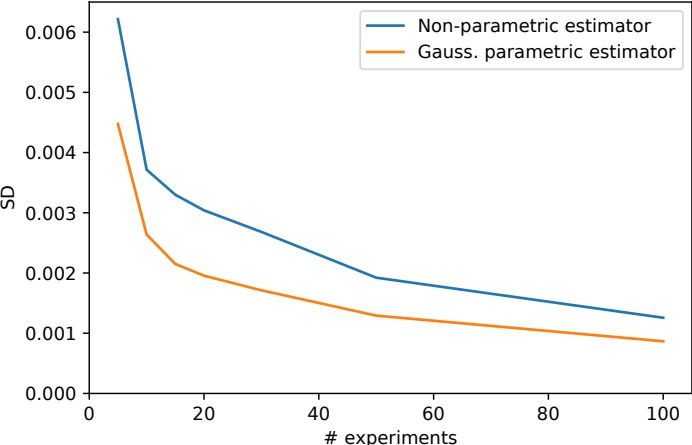

Figure 3: Empirical comparison of the variances of the non-parametric and Gaussian parametric Boo$_n$ estimators. The data come from experiments with the AS Reader on CBT with fixed hyperparameters.

# D. Dealing with Estimator Uncertainty

## D.1. Confidence intervals

If an estimator characterizing a performance distribution, say $\widehat{\text{Boo}_n}$ or average, is calculated from experimental observations, it is subject to random variation, so if another research team tries to reproduce the experiments, they generally get a different estimate. The more observations are collected, the more precise the estimate generally is. Confidence intervals provide a natural way to express this uncertainty. Their usage also gives a sense whether the number of performed experiments was sufficient to reduce the uncertainly to a reasonable level, which is again not frequently addressed in machine learning papers.

The construction of the confidence interval would be trivial if we knew the distribution from which our estimate was drawn (as opposed to the distribution of the performance!) – it is simply the interval between the appropriate quantiles, e.g. the 2.5th and 97.5th quantiles in the case of the $95\%$ confidence interval. Such distribution has been studied extensively for instance in the case of a mean of Gaussian random variables. However, in other cases, it is not known. If we know at least the distribution from which the individual observations were drawn, we can use Monte Carlo methods to precisely estimate the confidence interval; however, if we are not able to make an assumption about the underlying distribution, we need to use only what we have: our samples from the distribution. In such case the variability of our estimator can be approximated using the Bootstrap (Efron, 1979) or similar methods.

The Bootstrap consists of repeatedly sampling *with replacement* $m$ random observations from our pool of $m$ obser-

vations, say $B$ times. Each such sample is then used to calculate an estimate of our quantity of interest, say Boo$_n$ or mean. This creates a sample of $B$ values of the estimator. The confidence interval can then be easily estimated taking the appropriate quantiles from this resulting *Bootstrap* distribution of the estimator, which should be approximating the unknown underlying sampling distribution. The Bootstrap distribution has been shown to converge to the true underlying performance distribution.

If we know the underlying distribution (up to some parameters), we can estimate its parameters and then generate a simulated *Monte Carlo* sample from the distribution, which can be used to calculate a sample of the estimator and the corresponding confidence interval in a similar way as above with the advantage of the distribution being smoother.

Beside estimating the confidence interval for the value of Boo$_n$ or mean itself, either re-sampling method can be used to construct a confidence interval for the relative improvement of the newly proposed architecture compared to a baseline. The improvement can then be considered significant if zero is not included in the confidence interval. More details on constructing Bootstrap confidence intervals can be found in many standard texts on computational statistics, for instance in (Efron, 1987).

For illustration, we calculated the Bootstrap confidence interval for several sample sizes $m$ for Resnet and the AS Reader. Each was constructed using $B = 100,000$. The results are plotted in Figure 2.

# E. Comparison of estimators

Figure 3 shows the comparison of the non-parametric and Gaussian parametric estimators of $Boo_n$, both introduced in Section 3.2, in terms of their variance for various sample sizes. The parametric estimator shows a somewhat lower variance. This is an advantage if the performance distribution is indeed approximately Gaussian, which is the case for both cases with fixed hyperparameters that we tested in our experiments. However, this can introduce bias if the true performance distribution differs from the theoretical distribution assumed by a parametric estimator, so one should be prudent to use it.