# OpenReview forum: "A Boo(n) for Evaluating Architecture Performance"
_ICLR.cc/2018/Conference — Reject_

### Official Review · AnonReviewer3 · 2017-11-20
**A scalar measure for architecture performance**

**Rating:** 4
**Confidence:** 4

**Review:**

The authors propose a new measure to capture the inherent randomness of the performance of a neural net under different random initialisations and/or data inputs. Just reporting the best performance among many random realisations is clearly flawed yet still widely adopted. Instead, the authors propose to compute the so-called best-out-of-n performance, which is the expected best performance under n random initialisations.

Pros:
- The widespread reporting of just the best model is clearly leading to very biased results and does not help with reproducibility. Any effort to mitigate this problem is thus welcome.
- The proposed quantity is simple to compute if we have m realisations of the same model under different random inputs (random initialisation or random data) and will converge to a stable limit even if m is very large.

Cons:
- The best-out-of-n performance is well grounded if we have different random inputs such as random initial parameters or random batch processing. Arguably, there is even larger variance if the model parameters such as number of layers, layer size etc are varied. Yet these variations cannot really be captured by the best-out-of-n performance indicator unless modelled as random variables (which would lead to different sorts of problems).
- Computationally it requires to have a large number m of replications which is not always feasible.
- Most importantly: the proposed way is just one of many ways to reduce the distribution of performances to a single scalar quantity. Why is it better than just reporting a specific quantile, for example? Perhaps any such attempt to reduce to a single scalar is flawed and we should report the full distribution (or first and second moment, or several quantiles). For example: the boo-n performance gets better if the outcome is highly variable compared to a model where the mean performance is identical but the outcome much less variable. High variance of the performance can be negative or positive, depending on the application and the choice of boo-n is making a singular choice just as if we chose the mean or min or max or a specific quantile.

---

> ### Author Response · Authors · 2017-12-07
> **authors' response**
>
> - We address the first point in the common response above.
>
> - Yes, usage of the proposed method may require a large number of replications. However, this requirement stems from the degree of stochasticity in training. If we used any other statistical technique, we believe it would require a comparable amount of replications. If one really cannot afford to run so many replications, one should still try to estimate the resulting confidence interval and hence at least disclose the uncertainty in reported results. Today this uncertainty is still there, just unreported.
>
> - Yes, we agree that researchers should publish as much information as possible about the performance distribution of their architecture, which may allow the reader to calculate the characteristic that interests her the most (whether it be mean, Boo_n, or a quantile). However, we believe that scalar metrics do have their value as proxies for comparing models - this usage now has an important place in Machine Learning research. This is why we are trying to propose an improvement in this area.
> As to why we consider Boo_n better than the alternatives (e.g. mean, quantile), we believe that it best captures what may interest a practitioner intending to deploy the model: He may have the capacity to train n models and deploy the best one. Our score directly captures what performance to expect under such scenario.

---

### Official Review · AnonReviewer2 · 2017-11-24
**a step in the right direction**

**Rating:** 6
**Confidence:** 4

**Review:**

This manuscript raises an important issue regarding the current lack of standardization regarding methods for evaluating and reporting algorithm performance in deep learning research.  While I believe that raising this issue is important and that the method proposed is a step in the right direction, I have a number of concerns which I will list below.  One risk is that if the proposed solution is not adequate or widely agreeable then we may find a proliferation of solutions from which different groups might pick and choose as it suits their results!

The method of choosing the best model under 'internal' cross-validation to take through to 'external' cross-validation against a second hold-out set should be regarded as one possible stochastic solution to the optimisation problem of hyper-parameter selection.  The authors are right to emphasize that this should be considered part of the cost of the technique, but I would not suggest that one specify a 'benchmark' number of trials (n=5) for comparison.  Rather I would suggest that this is a decision that needs to be explored and understood by the researchers presenting the method in order to understand the cost/benefit ratio for their algorithm provided by attempting to refine their guess of the optimal hyperparameters.  This would then allow for other methods not based on internal cross-validation to be compared on a level footing.

I think that the fundamental issue of stochasticity of concern for repeatability and generalisability of these performance evaluation exercises is not in the stochastic optimisation search but in the use of a single hold-out sample.  Would it not be wise to insist on a mean performance (a mean Boo_n or other) over multiple random partitions of the entire dataset into training and hold-out?  I wonder if in theory both the effect of increasing n and the mean hold-out performance could be learnt efficiently with a clever experimental design.

Finally, I am concerned with the issue of how to compute the suggested Boo_n score.  Use of a parameteric Gaussian approximation is a strong assumption, while bootstrap methods for order statistics can be rather noisy.  It would be interesting to see a comparison of the results from the parametric and non-parameteric Boo_n versions applied to the test problems.

---

> ### Author Response · Authors · 2017-12-07
> **authors' response**
>
>
> - We admit that we add "yet another" method of evaluation. This does indeed create an opportunity for cherrypicking. However, we believe it has value to expand the pool of options in a situation where a large part of the community has settled on a standard we consider flawed. The situation may temporarily become messier, but we consider this a necessary step towards a new equilibrium, hopefully with better evaluation standards.
>
> - We will add the parametric/non-parametric estimator comparison to the paper soon.
>
> We add some additional remarks in the common response above.

---

### Official Review · AnonReviewer1 · 2017-11-30
**This paper introduces a metric designed to address the inadequacy of best model performance numbers but, since this metric requires many runs on the test data and also claims to normalize different experimental processes without a solid basis for this normalization, it may actually compound the problems highlighted.**

**Rating:** 4
**Confidence:** 4

**Review:**

This paper addresses multiple issues arising from the fact that commonly reported best model performance numbers are a single sample from a performance distribution. These problems are very real, and they deserve significant attention from the ML community.  However, I feel that the proposed solution may actually compound the issues highlighted.

Firstly, the proposed metric requires calculation of multiple test set experiments for every evaluation. In the paper up to 100 experiments were used. This may be reasonable in scenarios where the test set is hidden, and individual test numbers are never revealed. It also may be reasonable if we cynically assume that researchers are already running many test-set evaluations. But I am very opposed to any suggestion that we should relax the maxim that the test set should be used only once, or as close to once as is possible. Even the idea of researchers knowing their test set variance makes me very uneasy.

Secondly, this paper tries to account for variation in results due to different degrees of hyper-parameter tuning. This is certainly an admirable aim, since different research groups have access to very different types of resources. However, the suggested approach relies on randomly picking hyper-parameters from "a range that we previously found to work reasonably well". This randomization does not account for the many experiments that were required to find this range. And the randomization is also not extended to parameters controlling the model architecture (I suspect that a number of experiments went into picking the 32 layers in the ResNet used by this paper). Without a solid and consistent basis for these hyper-parameter perturbations, I worry that this approach will fail to normalize the effect of experiment numbers while also giving researchers an excuse to avoid reporting their experimental process.

I think this is a nice idea and the metric does merge the stability and low variance of mean score with the aspirations of best score. The metric may be very useful at development time in helping researchers build a reasonable expectation of test time performance in cases where the dev and test sets are strongly correlated. However, for the reasons outlined above, I don't think the proposed approach solves the problems that it addresses. Ultimately, the decision about this paper is a subjective one. Are we willing to increase the risk of inadvertent hyper-parameter tuning on the test set for the sake of a more stable metric?

---

> ### Author Response · Authors · 2017-12-07
> **authors' response**
>
> Regarding your concern with multiple test set evaluations: Yes, there are risks associated with it; however, we do not think using Boo_n would significantly change the current situation, which already mostly relies on the honesty of researchers - this is to some extent unavoidable in science.
> Even with Boo_n the researchers still can keep the test scores hidden aside and calculate the final Boo_n score only when they finished tuning the architecture and running the experiments (or even run evaluations with trained models only in the end).
>
> More importantly, we argue in the paper that reporting the test performance of a single model does not have that much scientific value. It's just a single sample drawn from a distribution, so we think it is not an appropriate way to characterize the performance distribution.
>
> To some extent, this draws on a distinction between doing science and simply competing on a challenge, e.g. on Kaggle. In the latter case, we should focus on fair conditions for competitors and your objections would be very appropriate. In the former case, we should mainly try to well characterize the behaviour of the model on which the researchers are publishing their findings. There, we believe that multiple test evaluation brings better insight.
>
> We address hyper-parameter tuning in the common response above.

---

### Author Response · Authors · 2017-12-07
**Response common to all three reviews: We are not primarly trying to compensate for hyperparameter tuning.**

Dear reviewers,

thank you for your insights. We all seem to agree that the problem of evaluation methodology is important. What your objections point to are flaws in our particular solution. We certainly admit that the solution is not flawless; however, we would certainly see purpose in starting a discussion on this important topic, which is largely absent from Deep Learning conferences while hundreds of papers continue reporting only the best single model performance, which is clearly inappropriate. 'Boon' is the best solution we were so far able to come up with, so we consider it a step towards improving the current situation.

Let us now address some of the issues you pointed out. Firstly, the method does not appropriately account for the effects of hyperparameter tuning. That is true and we probably didn't point it out clearly enough in the paper. What it was primarily developed to address was the problem of stochastic variation due to random initialization and data shuffling - that is randomness which appears in repeated training with fixed hyperparameters. In the paper, we show that this problem alone is non-negligible.
If we pick hyperparameters randomly, it produces the same effect and hence the problem can be solved using the same method.
However, normally, hyperparameters are chosen using either some Bayesian optimization or manual tuning. The improvement of results due to using these techniques over more experiment runs is much harder to model and discount in the evaluation procedure. It would be useful if we were able to do so, however it is beyond the scope of this paper. Still, these techniques are supposedly used because they are considered more efficient than simple random search, hence our method to correct for random search could be considered a lower bound on the correction due to more advanced hyperparameter tuning.
As a takeaway for us, we will try to make it clearer in the paper which problems we are addressing and which we are not.

We respond to some more specific points below each review.

Finally, given the reviews, the paper in its current form may well be rejected. We consider this issue important and hence may want to try to publish a new version elsewhere. What approach would you find most sensible? We could simply publish an analysis of the problems of the current practice, which we do in the first part of this paper, possibly with a review of existing alternatives. Or do you think the direction of Boo_n is generally right and would you encourage further work in that direction (in which case which points would you consider crucial to fix?)?

Thank you for your opinions.

---

### Decision · Program_Chairs · 2018-01-29
**ICLR 2018 Conference Acceptance Decision**

**Decision:**

Reject

**Comment:**

The subject of model evaluation will always be a contentious one, and the reviewers were not yet fully-convinced by the discussion. The points you bring up at the end of your rresponse already point to directions for improvement as well as a greater degree of precision and control.